# Picoplankton nitrogen guilds in the tropical and subtropical oceans: From the surface to the deep

Juan Rivas-Santisteban[1], Nuria Fernández-González[1,2☉], Rafael Laso-Pérez[1,3☉], Javier Tamames[1]*, Carlos Pedrós-Alió[1]*

1 Systems Biology Department, Centro Nacional de Biotecnología (CNB-CSIC), Madrid, Spain,
2 Facultad de Ciencias Experimentales, Universidad Francisco de Vitoria, Pozuelo de Alarcón, Spain,
3 Biogeochemistry and Microbial Ecology Department, Museo Nacional de Ciencias Naturales (MNCN-CSIC), Madrid, Spain

☉ These authors contributed equally to this work.
* jtamames@cnb.csic.es (JT); cpedros@cnb.csic.es (CPA)

## Abstract

Ecological guilds quantify the incidence and extent of resource transformation functions, irrespective of the species involved. Therefore, tackling the microbial nitrogen guilds is key to our understanding of the oceanic nitrogen cycle, but quantitative estimates of guild contribution across varying depths and under specific environmental conditions have yet to be accomplished. In this study, we examine the main picoplankton guilds participating in nitrogen cycling within the low and mid-latitude ocean ecosystems, from the surface down to 4000 m, using data obtained from 75 samples belonging to 11 stations in the Malaspina dataset. In particular, we used a quantitative approach to investigate the stability of nitrogen acquisition and nitrogen-redox guilds separately. Our results showed that nitrogen acquisition guilds are more stable and redundant than nitrogen-redox guilds across depths and site specific conditions. For example, differential conditions such as nitrogen depletion and oxygen availability affected the two groups of guilds in different ways. These findings have implications for the understanding of global nitrogen fluxes and the biosphere's functional diversification.

## 1 Introduction

Earth's water bodies do not constitute a single biome, but rather a compendium of them, corresponding to variable characteristics such as salinity, temperature, access to nutrients, irradiance, or hydrostatic pressure [1–7]. Under such ever-changing conditions, how can we trace causality in the observed microbial functional responses (i.e. appearance, depletion and diversification of biochemical functions)?

Microbial functional responses to these variables have been largely explored [8–12], although, as to our knowledge, they have not yet been depicted from a quantitative framework such as ecological guilds. An ecological guild refers to a list of organisms exploiting the *same resource in the same way* across environments, regardless

**Data availability statement:** Raw reads are deposited in the European Nucleotide Archive (https://www.ebi.ac.uk/ena/browser/view/PRJEB52452). S1 File with the annotation reference trees and files is available from the Zenodo public repository (https://zenodo.org/records/14557548). Code for guild analysis is available from the GitHub repository (https://github.com/pyubero/microguilds).

**Funding:** Funded by project PID2019-110011RB-C31/-C32 of Agencia Estatal de Investigación, Spanish National Plan for Scientific and Technical Research and Innovation. R.L.-P. was funded by a Ramón y Cajal grant (RyC2021-031775-I) from the Spanish Ministerio de Ciencia e Innovación (MCIN/AEI/10.13039/501100011033) and the European Union («NextGenerationEU»/PRTR). Work supported by Ph.D. fellowship PRE2020-096130 from the Spanish Ministerio de Ciencia e Innovación and the European Social Fund.

**Competing interests:** The authors have declared that no competing interests exist.

of their phylogenetic relatedness [13]. membership in a guild is not a permanent character, and the contribution of a member needs to be quantified in each context. That is to say, while "species A" will always be considered "species A" (permanent, static), function X can be performed only under certain conditions and species, which fluctuate (dynamic). This framework can thus disentangle the contributions of predictor variables to the functional profile. To this end, we screen the functional responses to differential environments in two ways: (1) *how* the function has been carried out (incidence and number of unique protein variants, and how each of them acts on the substrate/s), and (2) *who* implemented the necessary proteins (taxon-aware effects).

In the present contribution our purpose was to analyze the microbial guilds involved in the N cycle in the oceans with the guild approach. The N cycle is essential to Earth's biogeochemistry, transforming inert N into forms that sustain life, regulate ecosystems, and impact climate. Biotic processes play a fundamental role in the allocation and rate of the N cycle [14]. The reasons for understanding microbial performance regarding N cycling are manifold: (i) there exist up to seven redox states in the range of possible N compounds in the ocean, all of which may be differently transformed by microbes [10,15]; (ii) they consist of both organic and inorganic N species [16]; (iii) N compounds are used in a variety of ways, both as an energy source and as an essential nutrient, and can sometimes serve as an electron acceptor under suboxic conditions [17]; (iv) there is variability in N-limitation among different areas; and (v) N compounds are differentially limiting, for example, $N_2$ is never limiting, whereas nitrate ($NO_3-$) frequently limits phytoplankton growth [15,17]. Thus, the oceanic N cycle depends on numerous actors yet, in turn, its proper functioning is critical for the maintenance of ecosystems.

We quantified microbial nitrogen cycle guilds within low and mid-latitude oceanic biomes using metagenomic data from the Malaspina 2010 (MP) expedition [9,18]. We classified the Malaspina metagenomes according to their depth, establishing three broad categories: epipelagic (0-200 m depth), mesopelagic (200-1000 m depth) and bathypelagic (1,000-4,000 m depth). This depth stratification is commonly used to study marine ecology, as these defined oceanic layers often exhibit similar physico-chemical characteristics [19,20]. However, geographic location is also crucial in defining equivalent oceanic physicochemical regions. Longhurst's provinces represent one effort to systematize these zones [21]. Therefore, we considered both depth and specific locations to study how they influence the abundance and diversification of oceanic microbial functions (Fig 1a).

Akin to our categorization of the Malaspina samples using depth-location criteria, we also classified the guilds related to the N cycle according to the ecological processes in which they are involved. We established the two most general categories: (1) acquisition of N compounds, and (2) redox transformation of the N species acquired from the environment. Regarding the first, it comprises transporters for different N compounds. The second includes key proteins for the transformation processes where the imported N compounds act as electron donors or acceptors. With this clear division between contexts and guilds, we described their interplay as inferred from the Malaspina metagenomes.

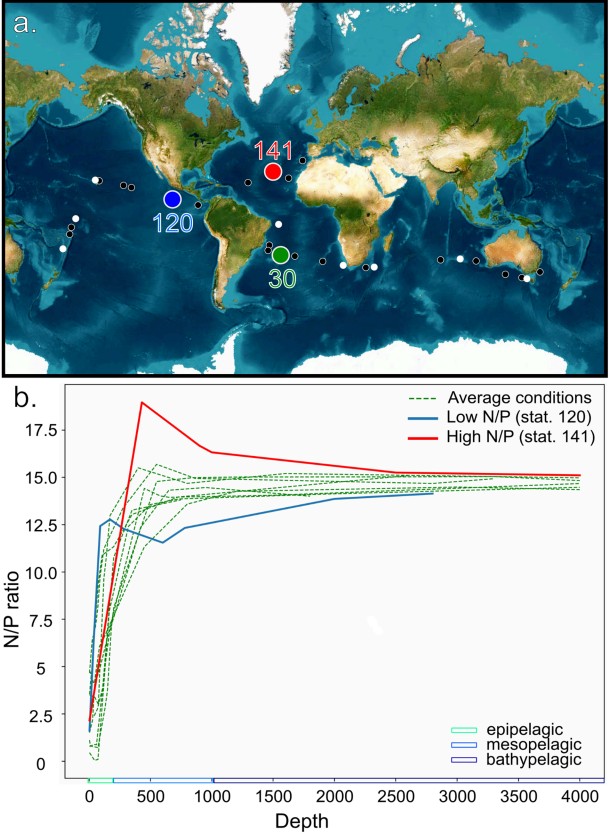

**Fig 1**. **a. Location of the samples and stations within the Malaspina dataset.** White circles represent the stations with samples along all the vertical profile, and thus the ones analyzed in this contribution. In green, station 30 in the South Atlantic is indicated. b. Values of N/P ratios (as inferred from $NO_x:PO_4$) with depth in the 11 Malaspina water columns sampled. Ratios were very similar at the epi- and bathypelagic but two stations showed marked differences in the mesopelagic: station 120 (Pacific, off the Mexico coast, with an oxygen minimum zone) and station 141 (North Atlantic, open ocean).

## 2 Results and discussion

### 2.1 N/P ratios of the main Malaspina water columns

In order to look for differences between the contributions of different guilds to the N cycle, we considered three depth intervals separately. We calculated N/P ratios to identify stations departing from the ocean average. This ratio provides a rough idea of how limiting is the dissolved N in an environment [22]. We assumed the ratios were causally linked to common nitrogen-related genes in various ocean regions. When we calculated this ratio in Malaspina samples (Fig 1b), the values showed less variability below mesopelagic depths, where they converged, being only marginally lower than the Redfield ratio (16:1), which is considered not to be either N or P limiting [23]. In effect, even though N/P vertical profiles were very similar in most stations, there were two that showed significant differences and we looked at them separately (Fig 1b). The first was a water column off the coast of Mexico (corresponding to station 120), where an Oxygen Minimum Zone (OMZ) occurs. This station had a very low mesopelagic maximum (N/P $\approx$ 12.4), and its values were always lower than the samples from other stations below 600m, as shown in Fig 1b. The second was a station from the open North Atlantic Ocean (corresponding to station 141), whose mesopelagic maximum was 18.96 and values were always higher than the other stations below this point (430m).

In summary, station 120 reached the mesopelagic N/P maximum faster with depth, close to $\approx$ 210m, while all the other stations showed the maximum at > 400m. Station 141, on the contrary, reached the mesopelagic maximum at the mean depth, but after that point it was never a nitrogen-scarce environment (N/P $\approx$ 16). Therefore, we selected these two water columns to study further changes in N guilds, as they showed opposite departures from the general pattern found regarding N depletion. We decided to analyze these stations separately.

We now have two criteria for selecting microbial functional responses: (i) depth, which is chosen because it covaries with other important factors (e.g., conductivity, salinity, temperature, oxygen, etc.; S1 Fig), (ii) N:P, which is chosen because we are investigating N-related guilds. Finally, we also chose to analyze the Station 30 in the South Atlantic (green) because, intriguingly, we detected anammox genes in an aerobic environment. Thus, we will show results for the average ocean (all other stations) and for each one of these three stations separately.

## 2.2 Taxonomic and depth distribution of N functional markers

We assembled the metagenomes, predicted the proteins, mapped the reads, and assigned the taxonomy for each marker gene as described in Methods. The full list of used marker genes and their associated pathways is available in S1 Table. Our annotation includes a step to purge false positives of the target function by using reference trees (for details see Rivas-Santisteban et al. [13]). Briefly, we build a tree with available sequences of a given gene and place the metagenomic reads in this tree. An example is shown in S2 Fig for the *hzsA* gene, a step in the anammox process. The tree has several different clusters. In particular the *hzsA-I* cluster includes all the know sequences belonging to *Planctomycetes* and thus, genes within this cluster are expected to carry out the function. On the other hand, sequences placed in cluster *hzsA-III* do not belong to any known anammox microorganisms and, further, we found them in an aerobic environment (SAtl station). Therefore, we discarded the sequences in cluster *hzsA-III*. We carried out the same procedure for every gene. The amounts of discarded sequences for nine genes are shown in S3 Fig. Some genes had almost no discarded sequences, for example *amt*. The *livJ* gene, on the other hand, had many discarded sequences. These differences depend on how well characterized is each gene and how similar are sequences of other genes that can be considered as paralogs. All the metagenomic queries (functionally annotated with abundances, filtered and unfiltered) used in this work are collected in S2 Table.

The abundance of the N functional markers (in normalized *features per million*, FPM) in the 75 samples appear in Fig 2). The taxonomic distribution for relative FPM *per gene per layer* is shown in Fig 2a–2c, so that the composition of the less abundant markers can be appreciated. The cumulative FPM is represented in Fig 2d–2e. The mean and standard deviation for the abundance of gene ORFs within each layer are shown in S3 Table. The most abundant genes were *amt* (marker for ammonia transport), *ilvC* (marker for branched amino acid synthesis), *nirK* (marker for denitrification or potential nitrification in archaea [24]), and *ureC* (marker for urea hydrolysis) regardless of depth. There was considerable species turnover between the epipelagic and mesopelagic, but the latter and the bathypelagic were similar. Primary producers (*Cyanobacteria*) and *Alphaproteobacteria* dominated at surface waters, while they were replaced by *Nitrososphaeria* with depth. As a consequence, the *nirK* gene became more abundant. The marker for N-fixation (*nifH*) was found in extremely few counts and it did not appear in the mesopelagic (Fig 2b and 2e). This is because most of the counts were filtered out as belonging to cluster IV of the *nifH* tree, which is widely considered non-functional [25] (S2 Fig). Finally, some functional markers were found profusely only at certain depths. For example, nitrate reduction (*nasA*) was three times more abundant in the bathypelagic than at other depths.

Overall, the most remarkable features of this distribution were: (1) that the most frequent genetic implementation for N uptake function in the oceans was the ammonium transporter; (2) that the metabolic capacity to build branched chain amino acids was very extended across oceanic layers and taxa (ubiquity of *ilvC*); and (3) that the main N redox process in the oceans involved *nirK* functions (denitrification/nitrification partial steps) only below the euphotic zone. These genes were mainly implemented by *Nitrososphaeria*. We are not aware of prior studies that have documented these specific

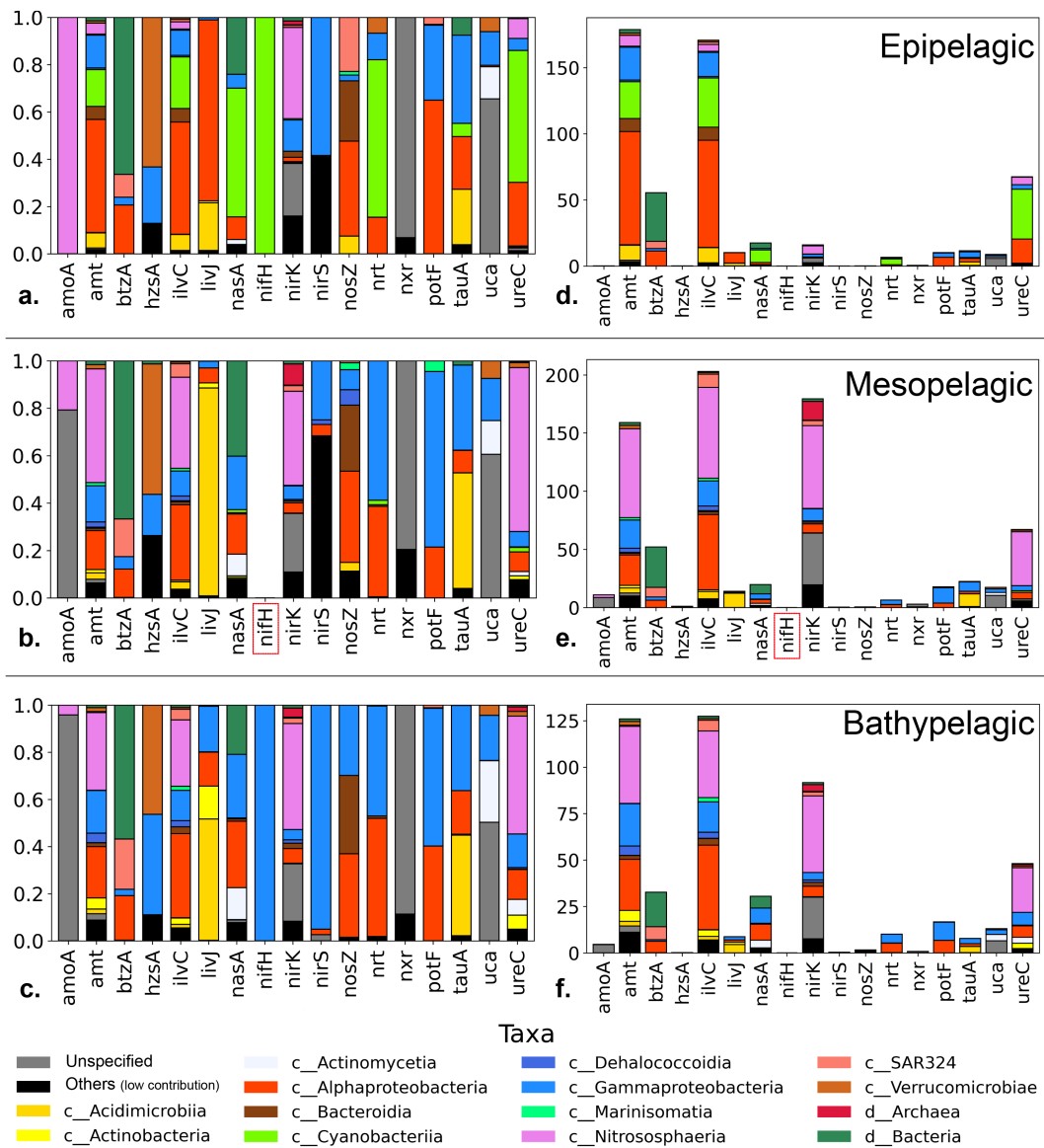

**Fig 2**. **Relative abundance of the studied genes in features per million (FPM) for the three water layers.** In panels d-f the total counts are shown. In panels a-c the percent taxonomic composition is shown to see the contributions of the less abundant components. Gene names are displayed in alphabetical order for easy location. The most abundant genes were amt (ammonia uptake), ilvC (carboxylated aminoacid uptake, nirK (denitrification) and ureC (urea degradation). The most important changes occurred between the epi and the mesopelagic, especially in the taxonomic composition of the most abundant genes, and in the abundance of amoA and nirK genes. The nifH gene for N fixation did not appear in the mesopelagic (indicated in red) and was very rare at other depths.

patterns. Regarding (1), it has been shown that *amt* copy number decreases with depth [26]. Our result is compatible as *amt* abundance slightly decreases with depth. Instead, here we show how, regardless of a drastic taxonomic change, the abundance of *amt* remains of the same order of magnitude.

Our results are, until now (Fig 2), analogous or complementary to other oceanographic studies retrieving the abundance of functional markers [9,27,28], with the difference of a careful annotation of metagenomic queries. However, to better describe the structure of the main N-related guilds, more details are required about these major functional actors,

in addition to their abundance; for example, the propensity of genes to remain, be depleted, or accumulate substitutions. This deeper knowledge may be approximated with the by-function genetic diversity (within a particular taxa) [13]. The following sections cover this matter in depth.

## 2.3 Diversification of N cycle proteins

One important quantitative aspect of a guild is certainly its abundance. If a given guild is twice as abundant in one sample as in another, this strongly suggest it is more important in the former sample. However, there is a second aspect. A guild could be represented by many identical sequences. For example, this could happen if there is a bloom of one taxon possessing the gene. On the other hand, the guild may be formed by many different sequences (even though able to carry out the function, naturally) irrespective of the total abundance. We postulate that this diversity of sequences within a guild is a relevant parameter since presumably higher diversity of sequences will provide stability, adaptability and resilience to the guild [29]. Our quantification method allows separating these two factors and to analyze them independently.

The density functions for $\delta$ values are shown in Fig 3. For any given sample (or collection of samples, for example the epipelagic samples of the OMZ station). The $\delta$ values of each taxon will be distributed along the x axis. When the observed diversity of proteins is equal to the expected diversity, $\delta$ will have a value of 1, and the corresponding data point will appear in the x-axis with a value of 0 (since the scale is logarithmic). If the observed diversity is higher than expected, the $\delta$ will have a positive value (displaced to the right of the 0) and when the observed diversity is lower than expected it will appear to the left. The accumulation of real points at each x value will result in a density function that approximates the distribution of $\delta$ values in the sample. The statistical method used to find and smooth this density function is the kernel density estimation. The result allows a visual comparison of the distribution of $\delta$ values among different samples (or collections of samples).

Let's look at the OMZ station first. In the case of the N uptake guilds the density functions for epi- and bathypelagic are fairly similar and centered around the value of $\delta$ = 1. This distribution is the most likely expected one. The curve for the mesopelagic, however, shows a displacement towards the left, indicating that many of the taxa showed a diversity of proteins lower than expected. On the other hand, this did not happen in the case of the N redox guilds.

In most cases, the curves were centered at the value of $\delta$ = 1, but with a certain dispersion of values towards the left: all the stations for N redox genes and NAtl and OMZ, SAtl for N uptake genes (but not the average ocean). The only exception to this distribution was the epipelagic from the SAtl station, that showed a density function displaced towards the right, indicating a diversity of proteins larger than expected from their abundance.

The general distribution, then, shows that most marine samples have most of their taxa with the diversity expected from their abundance, but in many cases, there is a subset of taxa with diversities lower than expected. In some particular cases, there is a clear departure from this general distribution. In the case of the OMZ bathypelagic, the N uptake genes showed lower diversity that expected. One possibility is that the absence of oxygen decreases the number of taxa that can live here. In the case of the epipelagic of the SAtl station, the higher diversity may be due to the presence of oil degrading bacteria such as *Alcanivorax*. This presence might be altered by natural oil seeps [30]. Indeed, station 30 is close to the Santos and Espirito Santo basins, where chronic seafloor oil seepage has been reported [31,32]. The latter would provide microbial communities with a large variety of organic compounds derived from hydrocarbons [31]. Perhaps the degradation of these compounds requires a larger variety of proteins with slightly different affinities.

The equivalent graphs considering the abundance of the functional genes only (and not the diversities) are shown in S4 Fig. In this case there were very few differences between N uptake and N redox genes. Also, the secondary maximum in most distributions was found to the right, not to the left as in the diversity graphs. And, finally, the main deviation from the general pattern was found again in the mesopelagic of the OMZ station. In this case, however, the secondary maximum was displaced towards larger abundances.

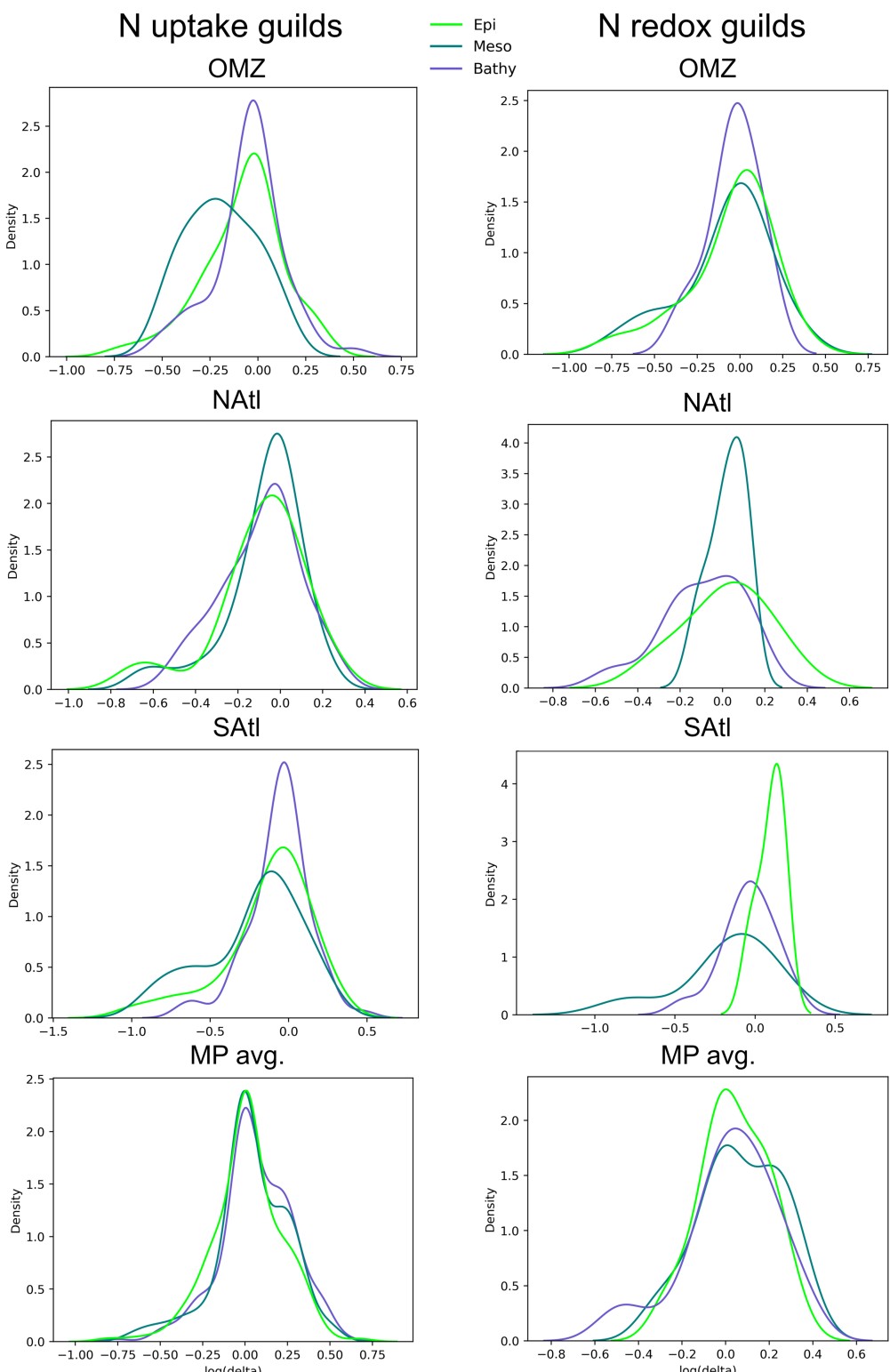

**Fig 3**. **Density functions for $\delta$ values for the different depths and stations analyzed.** Genes and taxa with a diversity of sequences expected from their abundance will have a $\delta$ of 1 and will appear in the x axis around the zero values (log(1) = 0). Those with a diversity lower than expected will appear to the left and, in contrast, those with a diversity larger than expected will appear to the right. In most cases the curves are centered around the value of zero. But in other cases there are clear departures, for example at the OMZ mesopelagic for N uptake genes.

It is apparent than considering abundance alone misses very interesting patterns. Having a diversity larger than expected from abundance suggests a guild with larger capacity for adaptation to different circumstances and, thus presumably, more stable in time and space. And this is independent of the abundance of the sequences in that guild at any given point.

In order to further quantify these relationships, we applied independent generalized linear models. The coefficients and their significance appear in Table 1. These models allow a statistical comparison between one set of data (for example the samples from the OMZ station) to the rest of the Malaspina dataset. Thus, we treat the predictor variables as categorical. The resultant coefficient provides an indication of the strength of the difference as well as the significance of such difference if it exists. As shown in Table 1, the NAtl and SAtl stations did not differ significantly from the ocean overage, neither in N uptake nor in N redox genes, while the OMZ station showed a rather significant negative relationship.

### 2.4 Description of oceanic N guilds

However, $\delta$ values ignore any surrogate indicator of the number of functional effectors readily available to operate in an environment. We thus quantified guilds using k-values (see methods). A full discussion of this metric can be found in Rivas-Santisteban et al. (2024) [13]. Briefly, these k-values include an estimate of the abundance of a given gene (A, number of features or reads of the gene) and an estimate of the unexpected diversity of sequences within such a gene ($\delta$ value). Generally, the diversity of sequences of a gene increases with its abundance. We show this relationship for both N uptake and N redox genes in S5 Fig. As shown, a relationship can be calculated for each gene from the environmental data providing the expected diversity ($d_{exp}$). Then, the actual diversity for a given gene in a given environment ($d_{obs}$) can be divided by the expected diversity. Thus, genes with an observed diversity equal to that expected from their abundance will have a $\delta$ value of 1. Genes with a diversity of sequences larger than that expected from their abundance are rewarded with a higher value of $\delta$, while those with a lower diversity are penalized. Then the $\delta$ value is multiplied times the abundance to obtain the k value. We do this because those genes with a higher diversity are expected to confer a higher stability and versatility to the guild under changing conditions. For a clear presentation, k-values were log transformed and displayed as radial plots (Figs 4 and 5).

**2.4.1 N uptake guilds.** The N uptake genes were divided between those incorporating inorganic N and those incorporating organic N. The marine microbiotas showed a general pattern in the relative abundances of these genes (Fig 4). Among the former, ammonia uptake (*amt*) was the most popular function, while the nitrate uptake gene (*nrt*) was always present but in much lower abundance. Finally, the nitrogen fixation gene (*nifH*) was almost below detection levels in all stations. This is in accordance with the energy required to reduce each compound to the redox level of organic matter: ammonia is already reduced, nitrate needs reduction to ammonia and nitrogen gas requires the very expensive N fixation process. The nitrate uptake gene was present at all depths but its abundance increased in the bathypelagic. The only exception to this pattern was in the mesopelagic of the SAtl station where *nrtA* found its largest abundance.

We had no previous hypothesis as to which of the organic-N uptake mechanisms would be more important. There are plenty of N-compounds in the oceans. We chose three genes responsible for the uptake of amino acids and three for

**Table 1. Generalized Linear Model of $\delta$ values for protein diversification.**

| Context | $N_{uptake}$ | | | $N_{redox}$ | | |
|---|---|---|---|---|---|---|
| | coef | std. err | P > |z| | coef | std. err | P > |z| |
| OMZ | −0.239 | 0.078 | 0.002 *** | −0.203 | 0.075 | 0.007 *** |
| NAtl | −0.105 | 0.078 | 0.178 | −0.0891 | 0.091 | 0.326 |
| SAtl | −0.1555 | 0.082 | 0.059 | −0.135 | 0.108 | 0.212 |
| Epi | 0.17 | 0.103 | 0.098 | 0.222 | 0.119 | 0.062 |
| Meso | 0.262 | 0.102 | 0.01 ** | 0.33 | 0.117 | 0.005 *** |
| Bathy | 0.432 | 0.103 | 0.000 **** | 0.268 | 0.118 | 0.023 * |

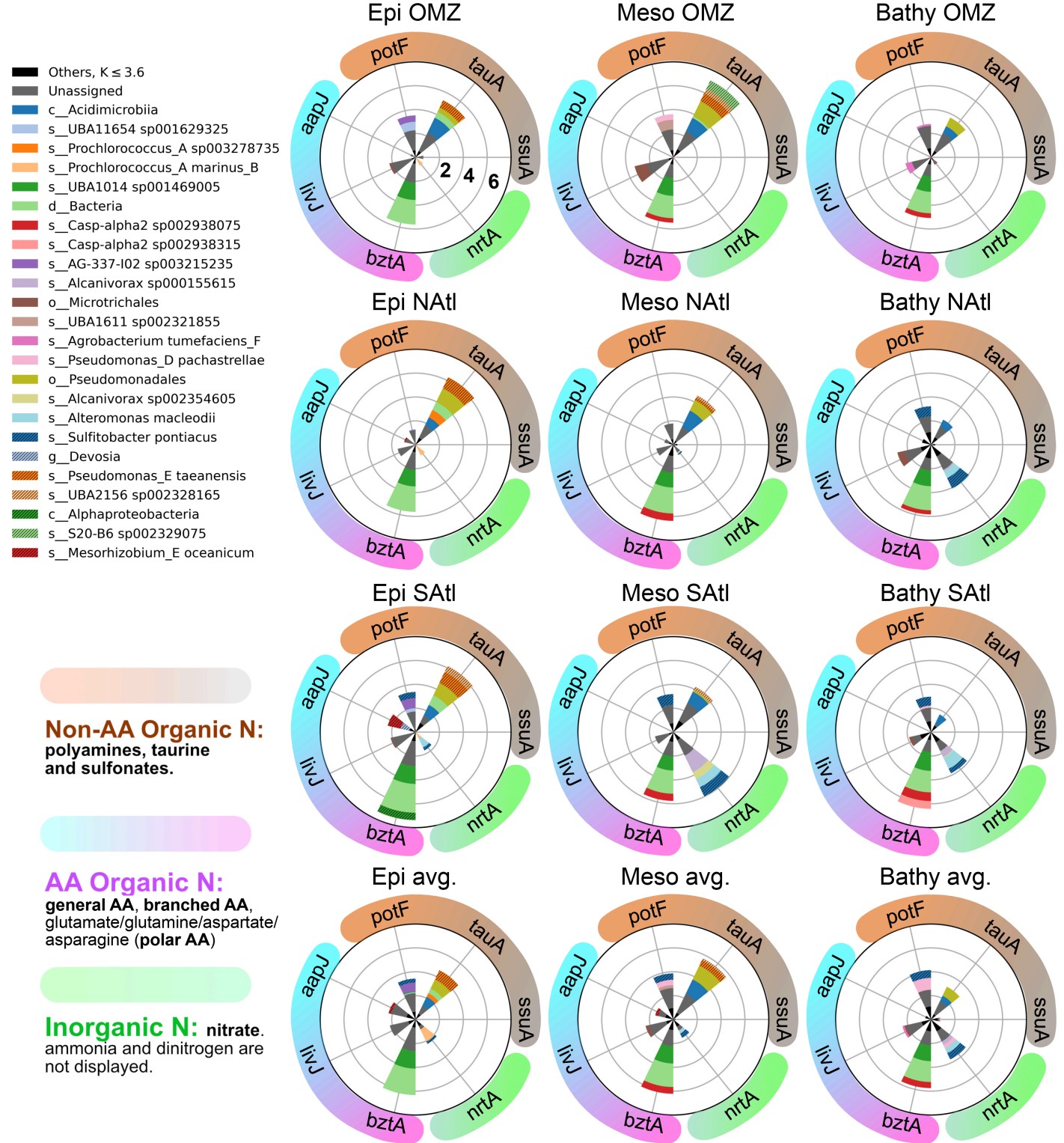

**Fig 4. Relative importance of the marker genes for N uptake guilds as represented by the log(k-values) that include components of abundance and unexpected diversity (see text).** We represented the top 25 taxonomic contributors, for the sake of clarity. All other taxa were grouped under Others (black). Genes have been organized into inorganic N uptake (green color, only nrtA, since amt is too abundant and nifH almost completely absent, and including them in the graphs would not allow visualization of most other genes), amino acid uptake genes (cyan to magenta), and organic non amino

acid uptake genes (brown shades). The concentric circles indicate orders of magnitude. The lower row of polar graphs shows the average for all stations except the three above. This shows the general pattern of N acquisition genes was similar in most stations. Although it is not shown, ammonia uptake was always important, as amt was always the most abundant gene. The three upper rows show values for the OMZ, the NAtl and the SAtl stations respectively.

other N-compounds. Among the former, the gene for the uptake of carboxylated amino acids *bztA* was the most abundant at all depths and stations. The other two genes (*livJ* for branched-chain amino acids and *aapJ*, a very general amino acid transporter) did change in abundance from one sample to another but they were always found in very low abundance compared to *bztA*.

Among the transporters of other N-compounds *ssuA* (transporter of sulfonates that have both nitrogen and sufur) was always very rare. The other two, *potF* for the transport of polyamines and *tauA* transporting taurine were fairly abundant in general but with differences. Gene *tauA* was more abundant than *potF* in the epi- and mesopelagic, while *potF* was more abundant in the bathypelagic.

These patterns were very robust since they appeared in almost all stations at all depths and, thus, they reflect the relative importance of the different N-compounds in the ecology of marine microbial communities. It is true that presence of a gene does not imply activity. However, the presence or absence of certain genes and not others is a strong indicator of their eco-evolutionary relevance [33,34].

**2.4.2 N-redox transformation guilds.** Nitrogen redox guilds were much more variable among depths and stations than nitrogen uptake guilds (Fig 5).

The potential for nitrification was present in most stations in the meso- and bathypelagic. Two taxa assigned to *Nitrosopelagicus* were the main contributors. There were two exceptions to this pattern. Neither station SAtl nor OMZ showed presence of the *amoA* gene. In the mesopelagic of the OMZ station oxygen was scarce and this is needed for nitrification. At the other depth intervals, however, there was plenty of oxygen. In OMZ station, the N/P ratio was the lowest found and, perhaps, this relative scarcity of nitrogen prevented nitrification from being an important process. The SAtl station, however, would require a different explanation. We will discuss this station below.

The OMZ station showed a very different pattern from that of the average ocean. Particularly in the mesopelagic, the depth of the oxygen minimum zone, had large abundances of both anammox and denitrification genes. This is reasonable since both processes take place in the absence, or at very low concentrations, of oxygen. The denitrification gene *narG* was present in all stations and depths, and its abundance generally increased from the epi- to the deeper sections. However, it was much more abundant in the oxygen minimum zone, as well as the other two genes examined (*napA* and *nosZ*).

Anammox genes were only found in the OMZ and, intriguingly, in the SAtl station. Thanks to our protocol to place metagenomic sequences in a phylogenetic tree of the available gene sequences, we found that the sequences from OMZ were placed in a clade with Planctomycetes-like *hzsA* sequences. Those from the SAtl station, on the other hand, appeared in another clade with no known anammox sequences (Figure S2). The latter were assigned to two *Alcanivorax* taxa. As mentioned, these waters were fully oxygenated and, thus, anammox was not expected. *Alcanivorax* is not known to carry out anammox or to have anammoxosomes. This gene (labeled *hzsA*-III) is a paralog that has probably acquired a different function. We do not have data to explain the proliferation of *Alcanivorax* at station SAtl, but it certainly changed the distribution of the nitrogen redox genes in these waters. Our *hzsA* metagenomic sequences were mostly assigned to *Planctomycetes*. However, there were several that were closer to those of *Verrucomicrobia*. The *hzsA*-I cluster built with data base sequences (S2 Fig) was mostly composed of *Planctomycetes*, but it also had sequences from *Verrucomicrobia* and even one assigned to a *Clostridium*, despite the fact that these bacteria have not been shown to carry our anammox or to have anammoxosomes. These assignments are likely the consequence of a poor characterization of the diversity of

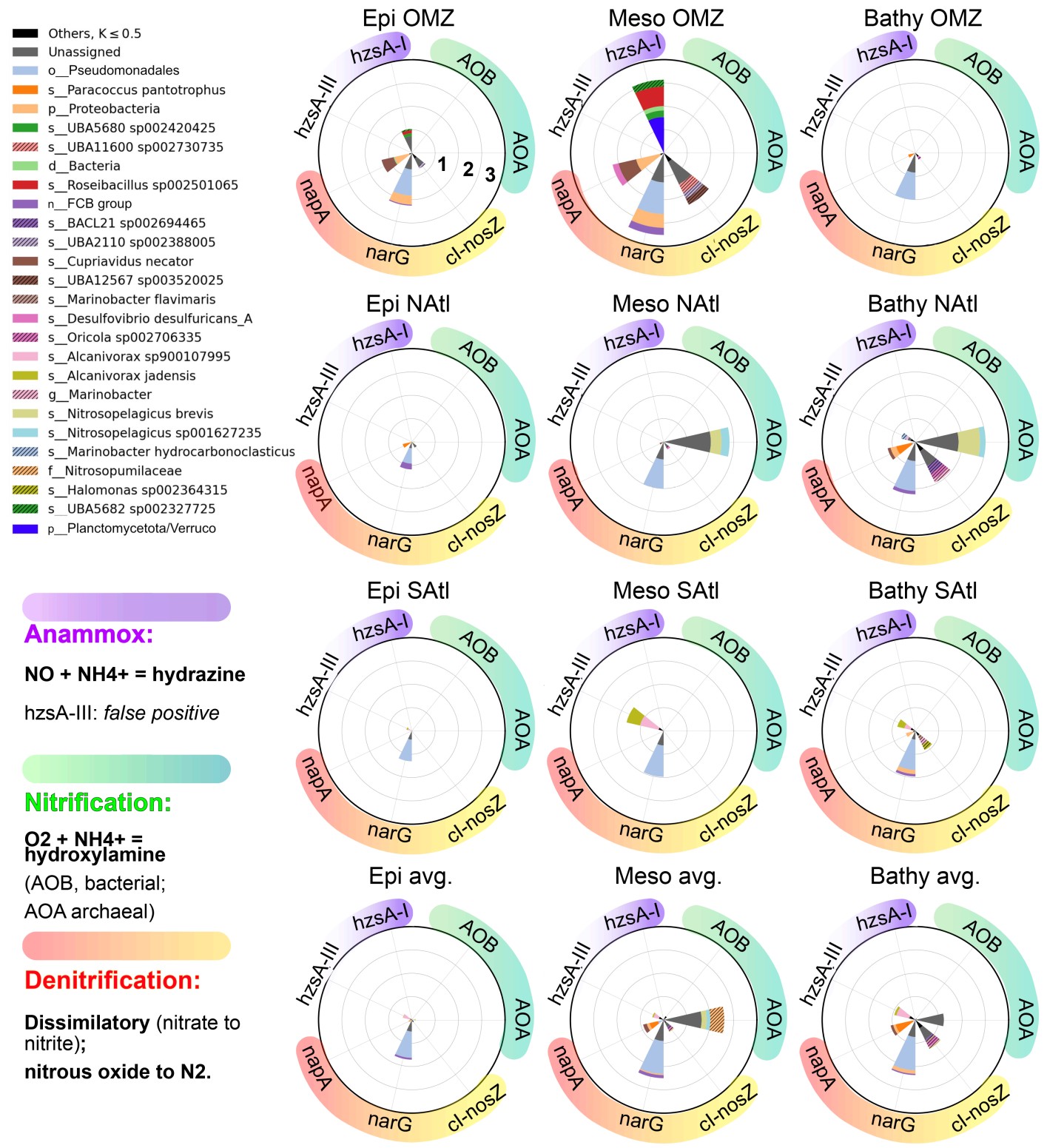

**Fig 5. Relative importance of the marker genes for N redox guilds as represented by the log(kvalues) that include components of abundance and unexpected diversity (see text).** We represented the top 25 taxonomic contributors, for the sake of clarity. All other taxa were grouped under Others (black). Genes have been organized into denitrification (pink to yellow shades), nitrification (green) and putative anammox (purple shades). Urea metabolism is not displayed. Differences among stations and depths were larger than for N uptake genes (see text).

*Planctomycetes* sequences in nature. Thus, we conclude that these sequences likely belonged to not well characterized *Planctomycetes*.

### 2.5 N:P ratios influenced picoplankton functional structure

**OMZ (station 120).** The mesopelagic signature at the low–N:P OMZ column matches established paradigms: denitrification and anammox are enhanced while nitrification weakens within the oxygen minimum [10,35,36]. Consistent with observations that aerobic ammonium and nitrite oxidation can persist at oxyclines and under transient nanomolar $O_2$ intrusions [37–39], nitrification features reappear above and below the core (Fig 6). What is distinctive in our data is the organization of these changes at the *guild* level. First, the strongest departures from the Malaspina average are concentrated in the mesopelagic redox guilds (Fig 6, left block), whereas N uptake functions keep a conserved pattern across depths. Second, a repeatable depth split in organic-N uptake is evident even in the OMZ column: taurine transport (*tauA*) is preferential in the epipelagic–mesopelagic. Third, while polyamine/putrescine transport (*potF*) is enhanced in the bathypelagic, in line with the role of organosulfonates and polyamines as labile DON substrates across the water column [40], we did find that putrescine-uptake guild in station 120 bathypelagic is diminished compared to other water columns. However, contrary to what has been reported [41], the taurine uptake guild is more prevalent beyond the epipelagic zone. Together, these patterns indicate that under low–N:P conditions the community shifts primarily along the redox axis, while access to nitrogen substrates remains buffered and widely shared in low N:P environments.

**NAtl (station 141).** In the high–N:P North Atlantic column, the tilt is the opposite: nitrification is comparatively stronger and N-loss pathways are depleted across depths (Fig 6, right block), consistent with the view that regional nutrient ratios covary with plankton biogeography and the balance of nitrogen transformations [22]. However we can now describe a reconfiguration of redox guilds with high N:P: First, the N uptake guild retains is much more variable than in low N:P conditions. Nitrate acquisition in st. 141 bathypelagic is prominent, while the sources of N are much more varied at this level. The occurrence of this phenomenon has been studied [9], but its strength was not linked to a higher N:P ratio. Second, nitrification and denitrification pathways (including dissimilatory) appear to be much more represented in the NAtl bathypelagic. Bathypelagic layer was described as mostly uniform regarding to its functional structure [9]; however, we here show that N:P ratios may be a local factor strongly influence the guilds therein.

### 2.6 The guild approach

The guild concept is particularly appropriate to analyze metagenomic data with functional and taxonomic information. A guild is composed of organisms that carry out the same function, with independence of being taxonomically related or not. The guild is responsible for the function presumably with different compositions in different environments. For example, urea degradation, as represented by the gene *ureC*, had the same relative importance at the three depth intervals studied (Fig 2d, 2e and 2f) but the organisms with this gene were very different at the epipelagic (*Cyanobacteria* and *Alphaproteobacateria*) from those at depth (mostly *Nistrosophaeria*).

This pattern was not only found with depth, but also among stations with different environmental conditions. An example of this is the guild represented by the *napA* gene, that codes the molybdoprotein subunit of the periplasmic nitrate reductase, involved in denitrification. In the OMZ station (Fig 4, upper row) the main contributor to this gene in the oxygen minimum zone (mesopelagic of station OMZ) was a *Desulfovibrio* strain, but it was absent from both the bathy- and epipelagic zones of this same station, where the guild was less important but still present. On the other hand, *Cupriavidus* was an important contributor to this guild in the epi- and the mesopelagic of the OMZ station. It was absent from the bathypelagic but it contributed again to this guild in the meso- and bathypelagic of the average ocean (Fig 4, lower row).

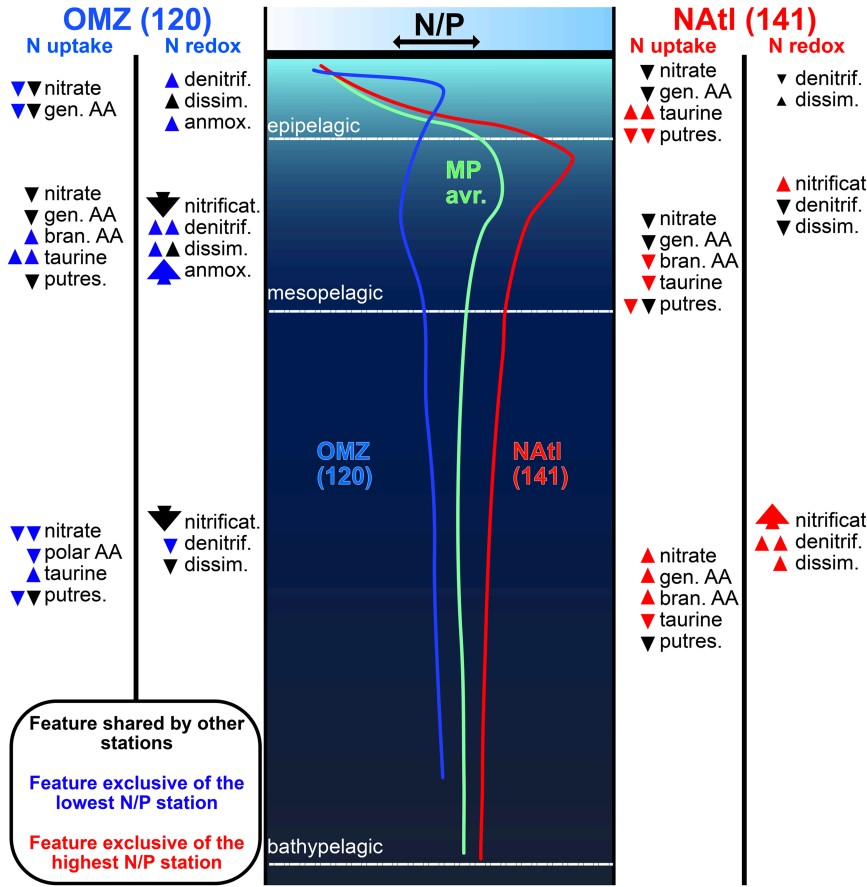

**Fig 6**. **Summary of the main findings.** The central cartoon shows the vertical profiles of the N/P ratios for the average ocean (green) and for the stations with the lowest (station 120, OMZ in blue) and highest (station 141, Natl in red) N/P ratios. The trends for the two later are shown at the left and right of the cartoon respectively. The relative strength of the change is shown qualitatively by the number and size of the arrows. The direction of the arrows (upward or downward) indicates whether the process in question increased or decreased with respect to the average. The black arrows indicate that this trend was found in at least another water column. When it was exclusively found only at one of these two stations the arrows are coloured, so it is more likely due to the difference in N:P. Abbreviations. Gen. AA: general L-amino acid incorporation. Bran. AA: branched amino acid incorporation. Putres.: putrescine uptake. Denitrif.: last step of denitrification + presence of the first steps of it. Dissim.: dissimilatory nitrate reduction. Anmox.: anammox.

These examples show clearly the uncoupling between taxonomy and function that usually occurs in microbial communities. In other words, functions cannot be easily predicted from taxonomic composition alone. Our quantification and graphical presentation allow an easy analysis of the two types of information together.

## 3 Conclusions

Our study has some limitations. First, Malaspina expedition only sampled tropical and subtropical waters but covered a substantial part of the Atlantic, Indian and Pacific Oceans. Thus, we cannot extrapolate to more temperate or polar waters. Next, the number of stations was also limited. However, the whole vertical profiles were analyzed down to 4000 m and our "average ocean" showed robust patterns. Finally, we only studied the picoplankton. Some functions, for example N fixation are known to be carried out in many cases by *Trichodesmium* aggregates or by symbionts of diatoms. We would have missed these genes. Despite this, diatoms are not generally abundant at the latitudes examined and our conclusions still hold for the picoplankton. Free-living diazotrophs are fairly rare in marine waters, accounting for up to 1% of the cells.

Even though their function is obviously important [42–44], our considerations about N uptake preferences by picoplankton are still real and consistent with the energy required to incorporate the different N compounds into organic matter. In summary, we found fairly robust patterns in the importance and distribution of certain N guilds in the tropical and subtropical oceans.

Despite departures at particular samples, general patterns in the distribution of the studied N guilds do exist in the ocean. These patterns included the following:

(i) The guilds for inorganic N uptake were in decreasing order of importance: ammonia uptake, nitrate uptake, and nitrogen fixation throughout the ocean in accordance with the energy required to process each one of them.

(ii) Among the guilds for organic N uptake, the gene for carboxylated amino acids *bztA* was the most abundant at all depths and stations. In the case of non amino acid N organic compounds *potF* for the transport of polyamines and *tauA* transporting taurine were fairly abundant in general. Gene *tauA* was more abundant than *potF* in the epi- and mesopelagic, while *potF* was more abundant in the bathypelagic.

(iii) N uptake guilds were less variable in importance than N redox guilds.

(iv) In the OMZ area examined, anammox genes were dominated by *Planctomycetes*.

(v) *Alphaproteobacteria* and *Cyanobacteria* were the dominant contributors to the N guilds in the epipelagic, while *Nitrososphaeria* dominated the N cycle in the meso and bathypelagic as shown by their contribution to N uptake (*amt*), denitrification (*nirK*) and nitrification (*amoA*), plus urea degradation (*ureC*).

(vi) Deviations from these general patterns have been summarized in Fig 6 for the OMZ and NAtl stations as compared to the average ocean.

(vii) Finally, considering not only abundance, but also unexpected diversity of sequences provides additional insight into the composition and variability of nitrogen guilds. For example, bathypelagic realm was associated to an increased $\delta$, regardless taxonomic or functional trends (Table 1 and S7 Fig).

## 4 Methods

### 4.1 Sampling and sequencing

A description of the Malaspina cruise can be found in Duarte (2015) [18]. Raw reads, full sampling and sequencing methodology can be found for example in Villarino et al. (2022) [45] and Sánchez et al. [46]. These are briefly summarized here. During the Malaspina 2010 expedition, conducted from 2010 to 2011, genetic and environmental data were gathered from 37 sampling stations spanning various oceanic regions, including the tropical and subtropical zones of the Atlantic, Indian, and Pacific Oceans, at depths ranging from 3 to 4000 meters. Sampling was done with Conductivity-Temperature-Depth (CTD) profilers with Niskin bottles from the surface to 4,000 meters, analyzing them for physical, chemical, and biological parameters.

All the samples from Malaspina (116) were filtered in two size fractions: small (0.2 $\mu$m) and large (3 $\mu$m) using 142 mm polycarbonate membrane filters. The small fraction comprised free-living bacteria, archaea, and picoeukaryotes, while the large fraction contained larger eukaryotes and particle-attached microorganisms. This study focused solely on samples from the small fraction. The majority of samples originated from the Atlantic, Pacific, and notably the Indian Ocean, where many physicochemical parameters were documented for the first time during the expedition. Given the tropical and subtropical nature of the samples, seasonal climate variations are expected to have a limited impact. Metadata included date, latitude, longitude, depth, temperature, filter size, conductivity, oxygen levels, fluorescence, Photosynthetic Active Radiation (PAR), Direct Photosynthetic Active Radiation (SPAR), turbidity, and salinity. Bottle depths and other details are indicated in S4 Table.

DNA sequencing was conducted using Illumina MiSeq with a 2 x 250 bp reads configuration, yielding a total read count ranging from 319 to 508 million reads per sample. Initial read processing involved contaminant scanning (e.g., Illumina

adapter sequences) and quality-based cleaning using FastQC [47], FastX-Toolkit [48], and SolexaQA [49] software. Trimmed reads falling below 45 nucleotides and quality score 33 were subsequently discarded.

## 4.2 Metagenomic workflow

All relevant first steps in the bioinformatic workflow were conducted with SqueezeMeta v.1.5.1 [50] predefined parameters. We assembled 75 metagenomes in sequential mode, retrieving individual genomes through binning procedures using DASTool software [51]. We limited the contig size to 200 bp with megahit [52]. Functional annotation was done by homology search with DIAMOND v.0.9.13 [53] against the NCBI database for taxonomic assignment and KO (KEGG Orthology) and COG databases for automatic functional assignment.

 Abundance estimation was automatically calculated with SqueezeMeta v.1.5.1. We provided gene abundances as FPM or *features per million* (a neutral term for TPM that are not derived from a transcriptome), recalculated globally on the sum of raw reads from the 75 samples that are the subject of this study. Thus, the sum of all functional features across samples yield 1 million FPM.

## 4.3 Functional delineation of metagenomic queries related to nitrogen cycling

To explore nitrogen cycling, we established reference sequence databases for 18 selected genes encoding proteins strictly involved in key steps of the nitrogen cycling (S1 Table), covering N redox functions (*amoA*, *nifH*, *nirK*, *ureC*, *uca*, *nxrA*, *nirS*, *nasA*, *narG*, *nosZ*, *hzsA*), nitrogen transporters—substrate-binding proteins (*amt*, *nrtA*, *tauA*, *potF*, *livJ*, *bztA*), and biosynthetic pathways (*ilvC*). These sequences include well-studied sequences with biochemical evidence, so that truly functional sequence spaces can be subsequently matched with short metagenomic queries. For this purpose, we reconstructed reference phylogenetic trees with the IQ-TREE software [54] for the above retrieved sequences, as per Rivas-Santisteban et al. [13]. These are available in File S1. We built these reliable trees as metagenomic sequence classifiers, to discern true markers from similar paralogs. The candidate genes were retrieved from metagenomes using the pertinent Hidden Markov Model [55] (PFAM, COGs, and/or KEGG) for each gene. Then, we identified truly-functional sequences based on the propagation of the functional annotations onto the reference trees. These candidate sequences were placed in the reference trees to distinguish particular sequences with biochemical evidence. We tagged the trees and placements with MetaTag v.0.1.1; [56]. True marker genes were retained for the functional analysis, and their coverage and abundance were estimated as depicted in 5.2.

## 4.4 Assessment of protein diversification

We assessed the expected sequence richness by fitting

$$d_{exp,s} = c_s A_s^{\gamma_s} \tag{1}$$

where $s$ were sequences considered for a taxon, implementation and environment, abundance was measured in *features per million* (FPM), richness was the natural number of unique sequences, and where $c_s$ and $\gamma_s$ were constants. Thus, $\delta$ was calculated as a coefficient between $d_{obs,s}$ (observed richness) and $d_{exp,s}$. Kernel density estimates of Fig 3 and S4 Fig were obtained with custom software. Additional statistical tests and details can be found in S6–S7 Figs and S5 Table.

## 4.5 Generalized linear model

This study employed a generalized linear model (GLM) to analyze the relationship between the predictor variables (contexts) and the response variable ($\delta$). Predictor variables were thus categorical. A GLM with a Gaussian distribution and an identity link function was specified for the analysis. This model was chosen based on the continuous nature of the response variable and the assumption of a linear relationship between the predictor and response variables.

 

The GLM was fitted to the data using the iterative reweighted least squares (IRLS) method. This method iteratively estimates the model parameters to minimize the residual sum of squares. The fit of the GLM was assessed using several metrics, including the log-likelihood, deviance, Pearson chi-square, and pseudo $R^2$ (Cragg-Uhler). These metrics were used to evaluate how well the model captured the variation in the response variable and to assess the overall goodness-of-fit. All statistical analyses were conducted with a significance level of $\alpha = 0.05$.

To report effects for every context (Table 1), we fitted the same Gaussian–identity GLM *repeatedly*, each time changing the reference so that a single *focal* category (e.g., OMZ, NAtl, SAtl; or Epipelagic, Mesopelagic, Bathypelagic) was contrasted against the rest. Concretely, we re-levelled the factor (or, equivalently, used a one-vs-rest indicator), refit the GLM, and extracted the coefficient, standard error, and *p*-value for that focal contrast. We then repeated this for each category and compiled the resulting estimates into Table 1. This rotation-by-reference procedure yields one coefficient per context without introducing interaction terms or overparameterizing a joint model. The reported effects are thus *marginal contrasts*: for example, the OMZ coefficient is the OMZ-vs-rest shift in $\delta$ (irrespective of depth), and the Bathypelagic coefficient is the Bathypelagic-vs-rest shift (irrespective of station).

## Supporting information

**S1 File. All reference trees and data needed for the various metagenomic placements, supplementary figures and tables.**
(ZIP)

## Acknowledgments

We thank P. Yubero and F. Puente-Sánchez for discussion.

## Author contributions

**Conceptualization:** Juan Rivas-Santisteban, Carlos Pedrós-Alió.

**Data curation:** Juan Rivas-Santisteban, Nuria Fernández-González, Rafael Laso-Pérez, Javier Tamames.

**Formal analysis:** Juan Rivas-Santisteban.

**Funding acquisition:** Javier Tamames, Carlos Pedrós-Alió.

**Investigation:** Juan Rivas-Santisteban, Carlos Pedrós-Alió.

**Methodology:** Juan Rivas-Santisteban, Nuria Fernández-González, Rafael Laso-Pérez, Javier Tamames, Carlos Pedrós-Alió.

**Project administration:** Javier Tamames, Carlos Pedrós-Alió.

**Resources:** Juan Rivas-Santisteban, Nuria Fernández-González, Rafael Laso-Pérez, Javier Tamames, Carlos Pedrós-Alió.

**Software:** Juan Rivas-Santisteban, Nuria Fernández-González, Rafael Laso-Pérez, Carlos Pedrós-Alió.

**Supervision:** Javier Tamames, Carlos Pedrós-Alió.

**Validation:** Juan Rivas-Santisteban, Javier Tamames, Carlos Pedrós-Alió.

**Visualization:** Juan Rivas-Santisteban, Javier Tamames, Carlos Pedrós-Alió.

**Writing – original draft:** Juan Rivas-Santisteban, Carlos Pedrós-Alió.

**Writing – review & editing:** Juan Rivas-Santisteban, Nuria Fernández-González, Rafael Laso-Pérez, Javier Tamames, Carlos Pedrós-Alió.

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
