## [Decision Letter · Decision Letter 0]

7 Aug 2025

PONE-D-25-08884Picoplankton nitrogen guilds in the tropical and subtropical oceans: from the surface to the deepPLOS ONE

Dear Dr. Alio,

Thank you for submitting your manuscript to PLOS ONE. After careful consideration, we feel that it has merit but does not fully meet PLOS ONE’s publication criteria as it currently stands. Therefore, we invite you to submit a revised version of the manuscript that addresses the points raised during the review process.

We look forward to receiving your revised manuscript.

Kind regards,

Judi Hewitt

Academic Editor

PLOS ONE

Journal Requirements:

“Funded by project PID2019-110011RB-C31/-C32 of Agencia Estatal de Investigación, Spanish National Plan for Scientific and Technical Research and Innovation. Funded by MCIN/AEI/10.13039/501100011033 from the Spanish Ministerio de Ciencia e Innovación and the European Union ("NextGenerationEU"/PRTR). Work supported by Ph.D. fellowship PRE2020-096130 from the Spanish Ministerio de Ciencia e Innovación and the European Social Fund.”

5. Please note that funding information should not appear in the Acknowledgments section or other areas of your manuscript. We will only publish funding information present in the Funding Statement section of the online submission form. Please remove any funding-related text from the manuscript. 

6. Please note that your Data Availability Statement is currently missing the repository name. If your manuscript is accepted for publication, you will be asked to provide these details on a very short timeline. We therefore suggest that you provide this information now, though we will not hold up the peer review process if you are unable.

7. We note that Figure 1 in your submission contain satellite images which may be copyrighted. All PLOS content is published under the Creative Commons Attribution License (CC BY 4.0), which means that the manuscript, images, and Supporting Information files will be freely available online, and any third party is permitted to access, download, copy, distribute, and use these materials in any way, even commercially, with proper attribution. For these reasons, we cannot publish previously copyrighted maps or satellite images created using proprietary data, such as Google software (Google Maps, Street View, and Earth). For more information, see our copyright guidelines: http://journals.plos.org/plosone/s/licenses-and-copyright.

1) You may seek permission from the original copyright holder of Figure 1 to publish the content specifically under the CC BY 4.0 license.  

2) If you are unable to obtain permission from the original copyright holder to publish these figures under the CC BY 4.0 license or if the copyright holder’s requirements are incompatible with the CC BY 4.0 license, please either i) remove the figure or ii) supply a replacement figure that complies with the CC BY 4.0 license. Please check copyright information on all replacement figures and update the figure caption with source information. If applicable, please specify in the figure caption text when a figure is similar but not identical to the original image and is therefore for illustrative purposes only.

8. We are unable to open your Supporting Information file [supp-1.zip]. Please kindly revise as necessary and re-upload.

9.If the reviewer comments include a recommendation to cite specific previously published works, please review and evaluate these publications to determine whether they are relevant and should be cited. There is no requirement to cite these works unless the editor has indicated otherwise. 

**Additional Editor Comments:**

It is important to note the differences between the 2 reviewers reflecting their different backgrounds.  Because you are addressing a subject crossing discipline boundaries it is important to make your writing clear to all disciplines.  The biodiversity debate is an obvious example of this and including some references as to the utility of an index that includes both abundance and species richness is probably needed if you think that using that rather than abundance and richness separately brings more to the discussion. Other than that- they both offer good points about making the manuscript clearer.

Reviewers' comments:

Reviewer's Responses to Questions

**Comments to the Author**

1. Is the manuscript technically sound, and do the data support the conclusions?

Reviewer #1: Partly

Reviewer #2: Partly

2. Has the statistical analysis been performed appropriately and rigorously?

Reviewer #1: Yes

Reviewer #2: I Don't Know

3. Have the authors made all data underlying the findings in their manuscript fully available?

Reviewer #1: Yes

Reviewer #2: Yes

4. Is the manuscript presented in an intelligible fashion and written in standard English?

Reviewer #1: Yes

Reviewer #2: No

5. Review Comments to the Author

Reviewer #1: Rivas-Santisteban et al.,’s paper approaches the distribution of picoplankton (mostly bacteria) in the ocean from the perspective of ecological guilds, defined as lists of organisms exploiting the same resource (in this case nitrogen) in the same way regardless of their phylogenetic relatedness. This approach, rarely seen in biological oceanography but very interesting, is explored using the Malaspina dataset, with 11 CTD casts across the tropical oceans ranging from the epipelagic to the bathypelagic realm. I enjoyed reading this paper and, to my knowledge, this could be a valuable piece of work for many oceanographers who are not familiar with the ecological guild approach. However, I have a few concerns about the oceanographic interpretation of the results that I think could be improved (and I hope my suggestions are somehow helpful) and some writing suggestions to make the paper more readable and impactful.

Major points

Vertical and horizontal mapping - A strong point of this piece of work is the vertical extent and the geographic variability. Figure 1) is however quite confusing. For example: what do the colours of the dots in Figure 1a mean? I would suggest removing the equator line and the text on the figure (which is not very helpful) and instead include only the circles for the used stations (ideally connected by the shiptrack). Station 120 seems a long way away from the Mexico coast and there is another station that looks closer so I recommend checking that there are no mistakes there. Station 141 is marked as North Atlantic but it’s clearly in the South Atlantic (is that the SAtl station mentioned in the text?). I also think that a crucial piece of information is the distribution of depths of the samples. The bathypelagic is way thicker than the other realms so is the density of the vertical sampling the same? This might also affect the interpretation of the diversity and relative importance of the marker genes (if the sample numbers are different for different realms). I would also recommend to use Figure 1a at its best and ideally have the average oxygen distribution in the background since it’s something used to interpret the results. The data should be freely accessible from WOCE (https://woceatlas.ucsd.edu/) or there are some other model re-analyses that could help. I’m also a bit curiousby a few coastal stations (like those off South Africa and South Australia) - how do they relate to the average?

North Atlantic station - The authors provide a reasonable explanation of why station 120 is different from most of the other locations, but I didn’t find the same level of detail for the North Atlantic station which still presents a very interesting pattern. I wonder if presenting in the supplementary information some profiles of temperature, salinity and oxygen can possibly help the interpretation?

More details are required for the GAM model - I think it is something that is really important in the logic of the paper but I struggled to understand. Is the predictive variable a categorical one? And are any interaction terms considered? For example how would the delta from a bathypelagic sample in the OMZ be treated?

Minor points

Line 99 - ‘as expected’ : why is that expected?

Line 101 - were any other stations in OMZ? Perhaps having some oxygen distribution map on the background of figure 1 could be really helpful.

Line 176: ‘could be divided’ I suggest replacing this with ‘were divided’

Lines 283-288: This is an interesting interpretation, but to be plausible I think more detail should be provided: How frequent are oil spills in the area? And for how long do we expect them to impact the bacterial community? This paragraph also needs to be backed up with some references

Line 311: I’d avoid using the term ‘general ocean’ as that could mean something different to different readers.

Section 4.1: Here I think more details about the bottle depths, etc need to be included.

Reviewer #2: In their study, Rivas-Santisteban and co-workers ask whether there are picoplankton “guilds” that are responsible for two aspects of nitrogen metabolism in the tropical and subtropical oceans – uptake (acquisition) and metabolism (redox). They do so by analyzing metagenomes collected on the Malaspina cruises from 75 depths, focusing on a selected set of genes representing these two types of processes. Additionally, they incorporate into their analyses not only the abundance of the different genes but also their diversity, reasoning that increased diversity also means that the functions (or the guilds performing them) are more robust to perturbations.

The questions asked by the authors are surely interesting, and, in principle, I think their analyses could answer these questions. I especially respect the care taken by the authors to make sure that the sequences they analyze are true orthologs, rather than paralogs (or just genes with a level of similarity above a specific cutoff). However, I found the manuscript extremely difficult to read and follow. Perhaps I do not have all of the ecological background needed to understand the manuscript, but the following examples can highlight some places where I found the manuscript to be very unclear. I therefore recommend it undergoes a major revision before being acceptable to PLoS One.

1) In the introduction, I struggled to understand what the authors mean by, for example, “functional response” (lines 37 and 38), “membership is not a static quality” (lines 42-43), what are the predictor variables and the functional profile (lines 43-44), and what is the “clear division between biomes and guilds” (line 93). It was also not clear what the research question was, whether such questions were asked before (e.g. as part of the set of papers from TARA oceans), and how the specific data and methodology would build upon previous knowledge.

2) I also found the concept of “expected richness” difficult to follow, both in the materials and methods (lines 436-442), in the supplementary information (Figure S3), and in the results section (lines 159-174). While I understand in very broad terms that the diversity of sequences is expected to increase with their abundance, there must be many cases where this is not so clear. Additionally, there are many different measures of diversity that are commonly used for sequences (e.g. alpha diversity estimates often used in microbial ecology for 16S or functional genes), and I need a much more detailed reasoning why a metric that includes both abundance and diversity (some sort of product of these different measures) is more helpful to understand the community. I recommend the authors re-structure their manuscript, starting with the abundance (and what can be learned from it), and only then introducing the importance of diversity, and where it contributes another layer to the understanding of the guilds.

3) The authors focus on set of genes (lines 420-422), but expect the reader to know what pathways these genes belong to. When the results are shown (e.g. in Figure 2), these genes are ordered alphabetically and not, for example, by function. I think a good first step would be to present these genes in an organized manner, perhaps as figure 1, explaining which pathway each gene relates to. Then, in subsequent figures, the genes could be organized in such a way as to make analysis of the functions more clear (as the authors did quite nicely in figures 3 and 4).

4) The summary figure is unclear to me, showing mostly functions that change in the water column between two individual stations and the mean of the others. I think these are things that have been studied intensively (certainly in terms of N cycling in OMZs), and I don’t think there is a detailed-enough comparison of the authors’ findings in this regard and previous literature. If the focus here is on guilds, perhaps a figure should be generated specifically asking which organisms do what, or which organisms are part of which “guild” (i.e. share genes) – and again, comparing these results with the literature.

Some additional specific remarks:

Abstract (line 23) and elsewhere – I don’t think there is any added gain by shortening “Malaspina” to “MP” (acronyms often only add confusion).

Lines 64-70 seem more relevant to the materials and methods.

Line 99, Figure 1: What do the authors mean by “N:P”? Is this NOx:PO4? NO3:PO4? TDN:TDP? Also, were all of the measurements used to calculate these ratios above the limit of detection of the instrument (especially at the surface)? I would also plot Figure 1B as a classical oceanographic depth profile (so depth on the Y axis, N:P ratio on the X).

Lines 98-101 – O2 could also affect the results (a “confounding factor”), but I don’t think the authors discuss this option much (perhaps I missed it?).

Lines 105-108 – how do these measurements of the N:P ratio compare with other studies of this ratio across the oceans? Could these be related, for example, to global deep water circulation patterns?

Lines 122-127 – I like that the authors are being careful with their data, and want to focus on “bona fide” sequences, but can they show that when lots of sequences are being removed (e.g. the hzsA-III clade) these sequences are not orthologs from a previously unknown group, rather than paralogs or just genes with similar sequences (e.g. due to common domains)?

Lines 150-155 – this is a nice interim summary. Has this been shown in the past (e.g. very high abundance of ammonium transporters)?

Line 380 – some citations seem to have a different format

Line 392 – Climate (seasonal) variation may have a smaller impact than in other regions but is it really “minimal” (e.g. in relation to N:P ratios)? I think some citations of seasonal studies in similar latitudes are needed to make this statement more robust.

Table 1 (and throughout the manuscript) – I’m not a statistical expert but I did not see many places where differences in abundance or diversity that were mentioned were associated with a statistical test. I realize that the authors have two specific depth profiles which they compare with “all the rest” but I think some sort of test should be made. Table 1 (the results of the GLM) seems to be limited to an overall comparison of the variation rather than specific genes or functions.

Figures 2 -4 – what are the letter followed by an underscore (e.g. “c_” or “d_”) at the beginning of the taxa names? How are the taxa organized in the two figures (what order)? In figures 3 and 4, could they be organized by, for example, the “guilds”, and then colored in a way that can help highlight this? Also, some of the taxa in Figure 3 are unclear (e.g. “Casp-alpha…”). What is the taxonomic rank (or ranks) shown?

Also in Figure 3 – where is Urea found (it is not in any of the groups – non-AA organic N, AA organic N and inorganic N).

Figure 5 – the black arrows show change compared to what?

6. PLOS authors have the option to publish the peer review history of their article (what does this mean?). If published, this will include your full peer review and any attached files.

Reviewer #1: No

Reviewer #2: No

---

## [Author Response · Author response to Decision Letter 1]

20 Sep 2025

these comments are already in the Response to Reviewer's document

---

## [Editor Report · Decision Letter 1]

8 Oct 2025

Picoplankton nitrogen guilds in the tropical and subtropical oceans: from the surface to the deep

PONE-D-25-08884R1

Dear Dr. Alio,

We’re pleased to inform you that your manuscript has been judged scientifically suitable for publication and will be formally accepted for publication once it meets all outstanding technical requirements.

Kind regards,

Judi Hewitt

Academic Editor

PLOS ONE
---

## [Editor Report · Acceptance letter]

PONE-D-25-08884R1

PLOS ONE

Dear Dr. Pedrós-Alió,

I'm pleased to inform you that your manuscript has been deemed suitable for publication in PLOS ONE. Congratulations! Your manuscript is now being handed over to our production team.

Kind regards,

on behalf of

Dr. Judi Hewitt

Academic Editor

PLOS ONE